# Dietary Intake and Diabetic Retinopathy: A Systematic Review of the Literature

**DOI:** 10.3390/nu14235021

**Published:** 2022-11-25

**Authors:** Janika Shah, Zi Yu Cheong, Bingyao Tan, Damon Wong, Xinyu Liu, Jacqueline Chua

**Affiliations:** 1Singapore Eye Research Institute, Singapore National Eye Centre, Singapore 169856, Singapore; 2Lee Kong Chian School of Medicine, Nanyang Technological University, Singapore 308232, Singapore; 3SERI-NTU Advanced Ocular Engineering (STANCE), Singapore 639798, Singapore; 4School of Chemistry, Chemical Engineering and Biotechnology, Nanyang Technological University, Singapore 639798, Singapore; 5Ophthalmology and Visual Sciences Academic Clinical Program, Duke-NUS Medical School, National University of Singapore, Singapore 169857, Singapore

**Keywords:** diabetic retinopathy, diabetic macular edema, diet, nutrition, nutrients

## Abstract

Diabetic retinopathy (DR) is a common microvascular complication of diabetes mellitus. The evidence connecting dietary intake and DR is emerging, but uncertain. We conducted a systematic review to comprehensively summarize the current understanding of the associations between dietary consumption, DR and diabetic macular edema (DME). We systematically searched PubMed, Embase, Medline, and the Cochrane Central Register of Controlled Trials between January 1967 to May 2022 for all studies investigating the effect of diet on DR and DME. Of the 4962 articles initially identified, 54 relevant articles were retained. Our review found that higher intakes of fruits, vegetables, dietary fibers, fish, a Mediterranean diet, oleic acid, and tea were found to have a protective effect against DR. Conversely, high intakes of diet soda, caloric intake, rice, and choline were associated with a higher risk of DR. No association was seen between vitamin C, riboflavin, vitamin D, and milk and DR. Only one study in our review assessed dietary intake and DME and found a risk of high sodium intake for DME progression. Therefore, the general recommendation for nutritional counseling to manage diabetes may be beneficial to prevent DR risk, but prospective studies in diverse diabetic populations are needed to confirm our findings and expand clinical guidelines for DR management.

## 1. Introduction

Diabetic retinopathy (DR; Figure 1) is a leading cause of vision loss globally. From 1990–2020, DR ranked as the fifth most common cause of preventable blindness and the fifth most common cause of moderate-to-worse visual impairment [1]. Approximately one in three people with diabetes mellitus suffer from DR and a third of these are afflicted with vision-threatening retinopathy, defined as severe non-proliferative DR or proliferative DR (PDR) or the presence of diabetic macular edema (DME) [2]. According to the Global Burden of Disease study, the age-standardized prevalence of blindness caused by DR showed a substantial increase between 1990 and 2020 in many regions of Asia [3], sub-Saharan Africa, as well as high-income North America [1]. The number of people with diabetes is estimated to be around 600 million by 2040 [4]. With this projected rise in the diabetic population coupled with increased life expectancy, the number of people with visual impairment due to DR is expected to rise worldwide [5]. Of concern is that DR is the most frequent cause of visual impairment among working-age individuals [1], and vision loss from DR places a considerable burden on patients’ quality of life (QoL) [6]. Therefore, finding effective ways to prevent or control the progression of DR is of critical importance.

Appropriate nutrition is an essential component of diabetes management [7]. Even though dietary guidelines for managing diabetes and prediabetes have been proposed [7], their role in the development and progression of DR has not been clearly defined. Nutrition counseling that works toward improving or controlling glycemic targets, attaining weight management goals, and enhancing cardiovascular risk factors (e.g., blood pressure, lipids, etc.) may benefit persons with DR. Studies show a favorable association between dietary changes and a reduction in the risk of DR [8,9]. Thus, adopting nutritional therapy in earlier stages may prevent the development and progression of DR and consequently help to reduce the treatment burden of this disease [10]. However, the risk factors for diabetes such as age, gender, and body mass index may not be necessarily risk factors for the development of DR. [11] Thus, the impact of diet modification on diabetes and that on DR may also differ.

Systematic reviews on the impact of diet on DR have been conducted [12,13,14,15,16]. Studies have recommended that the diet plays an important role in modifying the risk of DR by showing evidence of a protective effect of the Mediterranean diet, high fruit, vegetable, and fish intake, along with reduced calorie consumption, against the development of DR [12,13,15]. However, most of these dietary reviews on DR have focused on a specific food, nutrient, or dietary pattern [12,13,14]. Nevertheless, very few systematic reviews comprehensively assessed the entire spectrum of dietary components but are not very recent [15,16]. Several recent introductions of new dietary factors, i.e., selenium [17], vitamin B6 [18], vitamin B2 [17], choline [19], rice [20], cheese, wholemeal bread [21] and diet soda [22,23], with their influence on DR, are not included in previous comprehensive systematic reviews. For instance, two recent observational studies have highlighted diet soda as a risk factor for DR [22,23]. Additionally, more studies sharing information on the effect of already known dietary factors on DR are also available, thus adding more valuable knowledge to the nutritional impact on DR. For example, newly added observational studies showing the protective effect of tea [24] and Mediterranean food [25] on DR support a similar finding in a previous systematic review [15]. In contrast, the protective effect of the consumption of coffee [26], shown by a new observational study, was not seen in the previous systematic review [16,27]. Lastly, DME is a vision-threatening manifestation of DR, more commonly seen in severe stages of DR [28], and the association between diet and DME has not been reported in previous reviews.

In the present systematic review, we wanted to comprehensively summarize the current understanding of the associations between dietary components, DR and diabetic macular edema (DME).

## 2. Methods

### 2.1. Literature Search

Using the PRISMA checklist (Appendix A [29]; Figure 2), we conducted a systematic review of all studies published in peer-reviewed journals with no language restrictions. We retrieved articles from Embase, PubMed, Medline, and the Cochrane Central Register of Controlled Trials with a date range from January 1967 to May 2022. We systematically searched the database by combining the following keywords: diet OR dietary intake OR vitamins OR antioxidants OR nutrients OR fruits OR vegetables OR alcohol OR milk OR tea OR coffee OR carbohydrates OR fatty acid OR proteins AND diabetic retinopathy OR diabetic macular edema.

### 2.2. Study Selection

Our search methodology identified 5367 titles that were screened by ZY and systematically excluded if they did not meet predefined inclusion criteria. The exclusion was performed independently by ZY and vetted by JS, and uncertainty was clarified by JC. The reference list of those articles fulfilling the eligibility criteria was also verified for further relevant studies.

### 2.3. Inclusion Criteria

According to the PRISMA guidelines, a PICOS (participants, intervention, comparability, outcomes, study design) framework was used to formulate the eligibility criteria.

Participants—Studies including human subjects with type1, type 2 diabetic mellitus, or both.Study design—It included prospective, case–control, cross-sectional, and randomized controlled trials (RCTs).Interventions or exposure—Studies that evaluated dietary intake using tools such as validated food frequency questionnaires, 24 h dietary recall, dietary history, or general interviewer-administered questionnaires. Dietary intake components included specific food, beverages, micronutrients, macronutrients, and dietary patterns (Figure 3).Outcomes—It included prevalence, incidence, or progression of DR with or without DME. Studies that assessed DR outcomes by fundus photography, fundus examination using a direct or indirect ophthalmoscope, and fundus fluorescein angiography were accepted. Different scales for grading the severity of DR such as the Early Treatment Diabetic Retinopathy Study (ETDRS) and the International Classification system of DR were also accepted. The ETDRS is based on seven field stereophotographs, classifying DR from level 10 (absence of retinopathy) to level 85 (vitreous hemorrhage or retinal detachment involving macula). Conversely, the International Classification System grade cases into the categories of: no apparent retinopathy, mild, moderate, and severe non-proliferative retinopathy and final-stage proliferative diabetic retinopathy [30].

### 2.4. Exclusion Criteria

Animal studies, in vivo/in vitro studies and reviews.Studies that included the non-diabetic, pre-diabetic, or impaired glucose intolerance participants, or patients with special types of diabetes such as gestational diabetes.Studies with insufficient data, such as lack of exposure/outcome definitions or absence of statistical analysis which did not enable us to make conclusions.Studies that measured only biomarkers in serum, blood, or urine with no relation to dietary intake.Studies including intake in the form of supplements containing multiple different types of nutrients.Studies describing outcomes using abnormal retinal changes, microvascular complications, or visual acuity but not defined in the form of DR severity.

### 2.5. Data Extraction

Data on the name of the first author, year, type of study, sample size, diabetes type, and participant’s age were extracted for each included study. Data extraction also included the components of dietary intake, method of assessment of dietary intake, DR outcome, DR diagnosis and its classification, confounders adjustment, statistical analysis, and summary of key findings. The ZY author performed the data extraction which was vetted by the JS author, and the JC author clarified uncertainty.

### 2.6. Study Quality Evaluation

The modified version of the Newcastle–Ottawa Scale (NOS; Figure 4) was used to evaluate the quality of observational studies [31]. In brief, the NOS is a scoring system whereby a maximum of 9 stars can be awarded to each study based upon three main criteria [32]:Selection of participants (maximum of 4 stars).Comparability (maximum of 2 stars).Exposure (for prospective or cross-sectional designs) or outcome (for case–control designs) (maximum of 3 stars).

Studies were awarded an additional star if they incorporated validated methods to assess dietary intake like validated food frequency questionnaires (FFQs), 24 h dietary recalls, 3-day food records, or serum biomarker levels. Studies were categorized as low in quality when awarded <4 stars, medium for 5–7 stars, and high for >8 stars.

We applied the Cochrane Collaboration Risk of Bias tool to assess the bias risk in interventional studies, i.e., randomized controlled trials. Briefly, a study was considered to have an overall low risk of bias when all key criteria were graded as having low bias risk; overall medium bias risk when all key criteria were graded to have low or unclear bias risk; and overall high bias risk when one or more key criteria were graded to have a high bias risk [33].

## 3. Results

### 3.1. Description of Studies

We selected 54 papers from 4962 screened titles that met the requirements of our inclusion. (Figure 2). It included 3 interventional, 17 prospective, 29 cross-sectional, and 5 case–control studies.

### 3.2. Measurement of Exposures and Outcomes

Most observational studies measured the dietary intake using standard dietary methods such as 24 h recall (n = 4) [20,34,35], food frequency questionnaires (FFQ) (n = 23) [12,18,36,37,38,39,40,41,42,43,44,45,46,47,48,49,50], or 3-day food records (n = 3) [17,51,52]. A general-based interviewer-administered questionnaire was administered in 20 observational studies, and only one study evaluated dietary sodium intake from urinary excretion levels. Most of the studies assessed DR outcomes through fundus photograph (n = 30), 13 studies did through ophthalmology examination, or 5 studies from medical, clinical or hospital records, and 4 studies used a combination of photograph and examination (Table 1).

### 3.3. Methodological Quality

Of 51 observational studies, the majority had high NOS scores, with 37 classified as “high quality” (≥8 stars) and 14 classified as “moderate quality” (5–7 stars). Of the 3 interventional studies, 2 and 1 had a high risk and medium risk of bias, respectively (Table 1).

### 3.4. Relationship between Intake of Micronutrients to Diabetic Retinopathy

#### 3.4.1. Antioxidants

The association between carotenoids (n = 6), vitamin C (n = 5), Vitamin E (n = 6), riboflavin (n = 1), and selenium (n = 1) with DR is reflected in Table 2.

##### Carotenoids

Tanaka and associates conducted a prospective study, finding that carotenoid intake was associated with reduced incident DR using a multivariate cox regression analysis of (Q4 [8.4 mg/day] intake vs. Q1 [2.6 mg/day] intake, hazard ratio [HR]: 0.52, 95% confidence interval [CI]: 0.33–0.81, *p* < 0.01) [48]. Using a cross-sectional study design, Shalini and associates also found a beneficial effect of carotene in DR [36]. They found that the plasma concentration of both pro-vitamin A (PVA) carotenoids (α-carotene, β-carotene, γ-carotene, α-cryptoxanthin, and β-cryptoxanthin) and non-PVA carotenoids (lutein, zeaxanthin, and lycopene) was significantly lower in the DR group compared to no DR patients and healthy controls (*p* < 0.001) [36]. Similarly, Zhang and associates also showed that higher dietary intake of retinol (100 μg/day) in type 2 diabetes patients was associated with a lower risk of DR (odds ratio [OR]: 0.88, 95%CI: 0.79–0.98, *p* = 0.025) [38]. However, the remaining three cross-sectional studies did not find significant associations between carotenoids and DR [35,44,46].

##### Vitamin C

The relationship between vitamin C and DR has been controversial. A longitudinal cohort study by Tanaka and co-workers showed a protective effect of increased vitamin C intake on incident DR (Q4 [183 mg/day] vs. Q1 [67 mg/day], HR: 0.61, 95%CI: 0.39–0.96, *p* = 0.03) [48]. The work of Tanaka et al. was the only prospective study carried out on this topic. On the contrary, a cross-sectional study by Mayer-Davis and colleagues found an increased risk for more severe DR when vitamin C intake increased from the first quintile of intake to a higher level of intake. This result, however, is significant only for the ninth decile (OR = 2.21, *p* = 0.011) [35]. Prospective cohort studies measure events in chronological order and can be used to distinguish between cause and effect, whereas cross-sectional studies measure parameters at a single timepoint and do not permit distinction between cause and effect. Few other studies, however, suggest no association between vitamin C intake and DR before and/or after adjustment [17,46,50].

##### Vitamin E

The association between Vitamin E and DR remains uncertain. She and colleagues observed Vitamin E protective effects on DR (OR: 0.97, 95%CI: 0.95–1.00, *p* = 0.036) in their cross-sectional study after adjusting confounding factors [17]. Similarly, Granado-Casas showed a protective effect of Vitamin E on DR (OR: 0.85, 95%CI: 0.77–0.95, *p* = 0.006) [40]. Contrastingly, in a cross-sectional investigation by Mayer-Davis and colleagues, an increased intake of Vitamin E was associated with increased severity of DR among those not taking insulin (10th decile vs. 1st quintile, OR: 3.79, *p*< 0.02) [35]. The remaining one prospective and two cross-sectional studies did not report any significant association between Vitamin E and DR [46,48,50].

##### Selenium

A cross-sectional study conducted on the Chinese urban population by She and associates found selenium to have a protective effect against DR (OR: 0.98, 95%CI: 0.96–1.00, *p* = 0.017) [17].

##### Riboflavin

One cross-sectional study by She and associates found no significant difference between dietary intake of riboflavin in the DR group compared to the DR group (*p* = 0.129) [17].

#### 3.4.2. Vitamin D

Neither a prospective nor a case–control study found any significant association between dietary vitamin D intake and DR [43,45].

#### 3.4.3. Choline

A cross-sectional study by Liu and associates found that a higher dietary choline intake is associated with increased odds of DR in women compared with the lowest intake group (OR: 2.14, 95%CI: 1.38–3.31; *p* = 0.001) when using multivariable logistic regression models. However, this association was not statistically significant in men [19].

#### 3.4.4. Calcium

A case–control study by Alcubierre on the Spanish population found no significant association between dietary calcium intake and DR [45]. Still, their study had a small sample size (n = 283), and no adjustment of confounders was performed [45]. However, Chen and associates found a protective effect of increased dietary intake of calcium from the risk of DR (OR: 0.70, 95%CI: 0.54–0.90, *p* = 0.005) in their cross-sectional study on the Chinese cohort and adjusted for multiple confounders such as serum glucose, hemoglobin, and smoking status [34].

#### 3.4.5. Potassium

Chen and associates showed that increased dietary potassium intake was associated with reduced occurrence of DR (OR: 0.76, 95%CI: 0.59–0.97, *p* = 0.029) in their cross-sectional study [34], whereas Tanaka and colleagues did not find any significant association between potassium intake and the risk of DR in their prospective study [48].

#### 3.4.6. Sodium

The findings of a prospective study by Horikawa and associates indicated that, among patients who consumed less than an average of 268.7g of vegetables, high sodium intake was associated with a higher incidence of DR in elderly patients with type 2 diabetes (The results of third [4.4g/d], and fourth [5.9g/d] quartiles compared with the first quartile [2.5g/d], HRs were 2.61 [1.00–6.83], and 3.70 [1.37–10.02], respectively, *p* = 0.010) [37]. Another prospective study by Roy and colleagues reported increased sodium intake as a risk factor for DME progression (Q4 vs. Q1, OR: 1.43, 95%CI: 1.10–1.86, *p* = 0.008), but there was no significant association with DR. [49] The evidence provided by the remaining studies showed no association of sodium intake with DR [47,53,54].

#### 3.4.7. Vitamin B6

Horikawa and associates, using a prospective study design, reported that high vitamin B6 intake was associated with a lower incidence of DR in the Japanese population with type 2 diabetes (The Q4 [2mg/day] compared with the Q1 [0.9mg/day], HR: 0.50, 95%CI: 0.30–0.85, *p* = 0.010) [18].

### 3.5. Relationship between Intake of Macronutrients to Diabetic Retinopathy

#### 3.5.1. Fats/Fatty acids

Table 3 shows the association between monounsaturated fatty acids (MUFA; n = 9) and polyunsaturated fatty acids (PUFA; n = 8) with DR.

##### Monounsaturated Fatty Acids (MUFA)

A total of six studies evaluated the association between MUFA and DR. Out of these six studies, two were prospective studies, three were cross-sectional studies, and one was a case–control study (Table 3). Alcubierre and associates, who conducted a case–control study, reported that increased MUFA intake decreased DR prevalence (high MUFA intake [≥46.3g] vs. low MUFA intake [≤36.0], OR: 0.42, 95%CI: 0.18–0.97, *p* = 0.034) [42]. The cross-sectional study performed by Granado-Casas and associates also showed that intake of MUFA was associated with a lower frequency of DR (OR: 0.95, 95%CI: 0.92–0.99], *p* = 0.012) [40]. In contrast, Cundiff and associates showed an opposite relationship between MUFA intake and DR progression in their prospective study, but confounders such as HbA1c, duration of diabetes, or diabetes treatment were not adjusted [53]. The remaining studies found no significant relationships between MUFA intake and DR [46,49,52].

Oleic acid is a specific type of MUFA, and its influence on DR was evaluated by a total of three studies (one cross-sectional, one case–control, and one prospective study). A case–control study by Alcubierre and co-workers showed a protective effect of oleic acid from DR (highest intake [≥43.6] vs. lowest intake [≤32.2] OR: 0.37, 95%CI: 0.16–0.85, *p* = 0.017) [42]. A cross-sectional study by Granado-Casas and co-workers also reported a similar finding [40]. However, Roy and colleagues did not find any significant relationship between oleic acid and DR in their prospective study [49].

##### Polyunsaturated Fatty Acids (PUFA)

Sala-Vila and associates found that middle and older age type 2 diabetic patients strictly adhering to dietary long-chain omega-3 PUFA (LCω3PUFA) recommendation of at least 500mg/day was associated with a decreased risk of sight-threatening DR compared to those not fulfilling this recommendation (HR: 0.52, 95%CI: 0.31–0.88, *p* = 0.001) [12]. A cross-sectional study performed by Sasaki and colleagues found that among well-controlled diabetic patients, increased daily consumption of PUFAs was associated with a reduced likelihood of DR (OR: 0.18, 95%CI: 0.06–0.59), whereas an increased saturated fatty acid (SFA) intake was associated with an increased likelihood of DR (OR: 2.37, 95%CI: 1.15–4.88) [46]. In contrast, Cundiff and colleagues showed an increase in DR progression with a higher intake of PUFA, but adjustment for confounders were not performed [53]. The remaining three studies did not show significant relationships between PUFA intake and DR [42,49,52] (Table 3).

There are two interventional studies with contrasting results. One survey by Houtsmuller and associates found that subjects who consumed a diet of unsaturated fat, rich in linoleic acid, had a significant reduction in DR progression compared to those on a saturated fat diet (*p* < 0.01) [55]. However, Howard-Williams and colleagues assessed that participants compliant with a modified fat diet (high PUFA-to-saturated fat ratio) tended to have a lower incidence of DR than those on a low-carbohydrate diet (low PUFA-to-saturated fat ratio) [56]. Still, this difference was not statistically significant [56].

#### 3.5.2. Carbohydrates

A cross-sectional study by Granado-Casas, using adjusted multivariate analysis, showed that intake of complex carbohydrates was positively related to the presence of DR (OR: 1.02; 95%CI: 1.00–1.04, *p* = 0.031) [40]. On the other hand, two studies (one cross-sectional, one prospective) showed an inverse association between carbohydrate intake and DR progression, but neither study adjusted for confounders [52,53]. The other four studies using a multivariable-adjusted model found no significant association between carbohydrate intake and DR [41,42,46,49] (Table 3).

#### 3.5.3. Proteins

A prospective study by Park and colleagues found that the intake of glutamic acid and aspartic acid did not affect DR incidence [51]. Still, lower intake of aspartic acid showed an increased proliferative DR incidence, and the result remained consistent after adjustment (intake of aspartic acid in the highest tertile vs. lowest tertile for PDR, HR: 0.39, 95%CI: 0.16–0.96, *p* = 0.013) [51]. Another prospective study by Cundiff and colleagues showed that increased intake of proteins lowered progression of DR risk. Still, in their cross-sectional study, Roy and associates showed a risk relationship between protein intake and DR prevalence [52,53]. However, relevant confounders were not adjusted by these two studies. The remaining three studies, which adjusted for confounders, showed that dietary protein intake was not significantly associated with DR [42,46,49] (Table 3).

### 3.6. Relationship between Food Intake to Diabetic Retinopathy

#### 3.6.1. Fruits, Vegetables and Dietary Fiber

Increased fruit, vegetable and dietary fiber consumption was associated with reduced incident DR in a prospective study conducted by Tanaka and associates (fruits intake Q4 [225.4 g/d] vs. Q1 [21.5 g/d], HR: 0.48, 95%CI: 0.32–0.71, *p* < 0.01; fruits and vegetables intake Q4 [670.7 g/d] vs. Q1 [232.6 g/d], HR: 0.59, 95%CI: 0.37–0.92, *p* = 0.01; dietary fiber intake Q4 [19.7 g/d] vs. Q1 [9.6 /d], HR: 0.63, 95%CI:0.38–1.03, *p* = 0.07) [48]. For dietary fiber, one prospective and two cross-sectional studies reported a protective effect on DR [52,53,57]. However, three studies (two prospective and one case–control study) reported no significant associations [21,42,49] (Table 4).

#### 3.6.2. Rice

A prospective study by Kadri and associates found that rice consumption was significantly associated with DR occurrence (OR: 3.19, 95%CI: 1.17–8.69, *p* = 0.018) [20] (Table 4).

#### 3.6.3. Cheese and Wholemeal Bread

Consumption of cheese and wholemeal bread showed a reduction in the risk of DR progression among the working-aged Australian diabetic population (cheese intake highest quartiles vs. lowest HR: 0.58, 95%CI: 0.41–0.83, *p* = 0.007 and wholemeal bread HR: 0.64, 95%CI: 0.46–0.89, *p* = 0.04) in a prospective study conducted by Yan and colleagues [21] (Table 4).

#### 3.6.4. Fish

A prospective study by Kadri and colleagues showed that frequent fish consumption by diabetic patients reduced the risk of developing DR (OR: 0.42, 95%CI: 0.18–0.94, *p* < 0.05) [20]. Similarly, Chua and colleagues, using a cross-sectional design, showed that frequent fish consumption (>2 times/week) reduced the risk of DR progression (OR: 0.91, 95%CI: 0.84–0.99 per 1-unit increase in fish intake; *p* = 0.038) [39]. However, one cross-sectional study observed no association between fish and DR [21] (Table 4).

##### Fish oil

A prospective study by Sala-Vila and associates reported that consumption of two or more weekly servings of oily fish reduced the incidence of DR risk compared to those who did not consume this (HR: 0.41, 95%CI: 0.23–0.72, p < 0.002) [12]. In contrast, the association between fish oil intake and DR was found not to be significant by one prospective study [58] (Table 4).

#### 3.6.5. Other Types of Food

No association was seen between consumption of processed meat, breakfast cereal, and seafood and DR progression in a prospective study by Yan and colleagues [21] (Table 4).

### 3.7. Relationship between Beverage Intake to Diabetic Retinopathy

#### 3.7.1. Coffee

A cross-sectional study by Lee and associates showed that the consumption of ≥2 cups of coffee per day reduced the prevalence of DR (OR: 0.53, 95%CI: 0.28–0.99, p for trend = 0.025) and vision-threatening DR (OR: 0.30, 95%CI: 0.10–0.91, *p* for trend = 0.005) in the Korean diabetics less than 65 years of age [26]. However, in their cross-sectional study, Kumari and associates found no significant association between coffee and DR [59] (Table 5).

#### 3.7.2. Tea

Xu and associates found that long-term tea consumption (≥20 years) in elderly diabetic Chinese residents was a protective factor for DR compared to non-tea consumers (OR: 0:29, 95%CI: 0.09–0.97, *p* = 0.04) in their cross-sectional study [24]. Similarly, a case–control study on the Chinese diabetic population by Ma and associates reported a protective relationship between green tea intake and DR prevalence (intake vs. no intake, OR: 0.48, 95%CI: 0.24–0.97, *p* = 0.04) [60] (Table 5).

#### 3.7.3. Milk

No association was observed between milk and DR progression in a prospective study by Yan and colleagues [21] (Table 5).

#### 3.7.4. Diet Soda

Mirghani and colleagues, using a cross-sectional study design, found that diet soda (sugar-free carbonated beverage) consumption was associated with a higher risk of DR (*p* = 0.043) [22]. Another cross-sectional study by Fenwick and associates also found a positive association of diet drink (>4 cans [1.5 L]/week) consumption with proliferative DR (OR: 2.62, 95%CI: 1.14–6.06, *p* = 0.024) [23]. Still, no association was found between regular soft drinks and DR [23] (Table 5).

#### 3.7.5. Alcohol

A prospective study on Indians living in Singapore by Gupta and associates found that alcohol consumption was associated with a reduction in incident DR compared to non-drinkers (OR: 0.36, 95%CI: 0.13–0.98, *p* = 0.045). Among alcohol consumers, occasional drinkers (≤2 days/week) had reduced occurrence of incident DR (OR: 0.17, 95%CI: 0.04–0.69, *p* = 0.013) compared with non-drinkers [61]. The other studies, which also reported protective effect of light-to-moderate alcohol consumption on the prevalence of DR, were cross-sectional studies [62,63,64,65] (Table 5).

On the other hand, a cross-sectional study by Thapa and associates found alcohol consumption to be a significant risk factor for the development of any DR (OR: 4.3, 95%CI: 1.6–11.3, *p* = 0.004) and vision-threatening DR (OR:8.6, 95%CI: 1.7–47.2, *p* = 0.010) [66]. Similarly, a risk association was found between heavy alcohol intake and DR (heavy [>10 pints of beer/week] vs. none–moderate intake [<10 pints/week, RR: 3.5, 95%CI: 1.2–8.4, *p* = 0.02) in a prospective study by Young and associates [67]. A cross-sectional study also showed a risk association between alcohol and diabetic macular edema prevalence (*p* = 0.010) [68]. Three prospective studies, a case–control study, and a cross-sectional study did not find any association between alcohol consumption and DR [53,69,70,71,72] (Table 5).

### 3.8. Relationship between Broader Dietary Patterns to Diabetic Retinopathy

#### 3.8.1. Mediterranean Dietary Pattern

Ghaemi and associates reported a significant protective effect of the Mediterranean diet against incident DR in type 1 DM (OR: 0.32, 95%CI: 0.24–0.44, *p* < 0.001) and type 2 DM (OR: 0.68, 95%CI: 0.61–0.71, *p* < 0.001) in their prospective study [25]. An interventional study showed the benefit of consumption of the Mediterranean diet on reducing the incident DR (any Mediterranean diet vs. control diet, HR: 0.60, 95%CI: 0.37–0.96) in type 2 diabetics, when using a multivariable cox regression model [73] (Table 5).

#### 3.8.2. Total Caloric Intake

Two prospective studies by Cundiff (r = 0.07, *p* < 0.007) and Roy (OR:1.41, 95%CI: 1.15–1.92, *p* = 0.002) reported a risk associated between a high total caloric intake and DR progression [49,53] whereas Alcubierre and associates found no significant association between high caloric intake and DR in their case–control study [42] (Table 5).

## 4. Discussion

From our systematic review on dietary intake and DR, we found that intake of fruits, vegetables and dietary fibers, fish, Mediterranean diet, oleic acid, and tea beverages had a protective effect on DR. We also found that selenium antioxidant, vitamin B6, cheese, and wholemeal bread may have a protective effect on DR. Still, this outcome was based on only one study in each of dietary component. The consumption of diet soda, increased caloric intake, rice, and choline was found to be associated with a greater risk of DR. In contrast, no significant association was found between vitamin C, riboflavin, and vitamin D and milk with DR. Other dietary components such as carotenoids, Vitamin E, potassium, unsaturated fatty acids, carbohydrates, coffee, and alcohol showed no clear relationship with DR, signifying that more studies are needed. The assessment of the influence of dietary intake on DME is limited to only one prospective study. This study found that a high intake of sodium was associated with DME progression. The findings from our systematic review may complement the current dietary recommendations for managing DR.

### 4.1. Protective Associations between Dietary Intake and Diabetic Retinopathy

In our review, high levels of consumption of fruits, vegetables, and dietary fibers has revealed strong protective effects against the development of DR [48,52,53,57]. Fruits and vegetables are rich sources of fiber and antioxidant compounds [74]. Dietary fiber delays glucose absorption from the intestines, thus reducing postprandial plasma glucose levels [75]. It also reduces inflammation and oxidative stress, which are known to be involved in the initiation and progression of diabetes [74]. Thus, dietary fiber would reduce the risk of hyperglycemia and oxidative stress-induced DR [76]. Fish oil is a rich source of long-chain omega-3 polyunsaturated fatty acid (LCω3PUFAs), which reduces the risk of diabetes [77] and is found to have a protective effect on DR in our review [12,20,39]. The retina is rich in LCω3PUFAs, particularly docosahexaenoic acid (DHA), which has anti-inflammatory and anti-angiogenic properties [78,79] and experimental studies have shown the protective role of supplemental DHA or LCω3PUFAs against DR or neovascularization of the retina [80,81].

The Mediterranean diet is a centuries-old eating pattern consisting of plant-based foods such as fruits, vegetables, legumes, nuts, and whole grains. It also includes fish and olive oil and a low intake of red meat, red wine, and saturated fatty acids [82]. Our findings show the protective effect of the Mediterranean diet on DR. The anti-inflammatory and antioxidant compounds in the Mediterranean diet indirectly improve the peripheral uptake of glucose and reduce peripheral insulin resistance, and are thus proposed to have a protective effect in preventing diabetic microvascular complications [83]. Similarly, the protective role of Oleic acid against DR seen in our review is also proposed to improve peripheral insulin sensitivity. The two observational studies in the Chinese cohort in our review have shown the protective effect of tea on DR. However, results must be interpreted with caution, as these studies did not take other dietary factors such as fruits and vegetables into account [24,60]. Tea is one of the most consumed beverages in the world, and tea extracts are reported to have antioxidants and neuroprotective properties, improving insulin sensitivity, inhibiting ocular neovascularization and vascular permeability [84,85].

### 4.2. Adverse Associations between Dietary Intake and Diabetic Retinopathy

Two cross-sectional studies found diet soda to be a risk factor in the progression of DR. The proposed mechanism is an alteration of gut microbiota leading to inflammation, oxidative stress, and cardiometabolic states such as obesity, insulin resistance, and diabetes [86]. Another proposed theory is that the overconsumption of other food or beverages might occur due to subjects overestimating the calories saved by substituting diet beverages for sugar-sweetened drinks [23]. However, further longitudinal studies are required due to a small sample size of 200 participants [22], as well as a lack of an account of changes in diet drink, i.e. from regular soft drink to diet soft drink for lifestyle modification upon diagnosis of diabetes, which could overestimate the relationship between diet soda and DR in the study [23].

A prospective study in our review showed that increased rice consumption, which increased the total caloric intake, contributed to the increased risk of DR occurrence. A systematic review by Wong and associates found that high caloric intake increases the risk of DR [15,49,53]. Experimental and clinical evidence suggests that high caloric intake increases oxidative stress in diabetic patients, thus possibly increasing the risk of DR [87,88,89]. Interestingly, in our review, carbohydrates, one of the main contributors to total caloric intake, have shown no significant association with DR. Still, one cross-sectional study has shown a positive association with DR [40]. Despite a lack of substantial relationship with DR, it is crucial to monitor carbohydrate consumption to control postprandial hyperglycemia in patients with diabetes [90]. Thus, encouraging low-glycemic index and low-calorie meal intake may be favorable to prevent the occurrence and progression of diabetic microvascular complications [91,92]. The risk of choline causing increased DR risk for females needs further investigation by cross-sectional [19]. The literature has reported the adverse effect of choline and its metabolite, trimethylamine-N-oxide, by aggravating vascular endothelial cell dysfunction, oxidative stress, and inflammation, which are critical mechanisms of DR development [93,94].

### 4.3. No Significant Association between Dietary Intake and Diabetic Retinopathy

We did not find any significant association between antioxidants such as vitamin C, E, riboflavin, carotenoid intake and DR. This similar finding was also reported by Lee and associates [95]. However, in investigational studies, antioxidant supplementations inhibit oxidative stress and the development of DR [96,97]. Similarly, experimental studies have shown a beneficial effect of PUFA against the development of DR due to its anti-inflammatory and anti-angiogenic properties [81,98]. Still, the current review shows an inconclusive association. The studies in our review showing the associations of alcohol intake with DR risk have demonstrated contradictory results. Thus, our review could not confirm the protective effect of alcohol against DR, which supports the meta-analysis by Zhu and associates [99]. A moderate amount of alcohol consumption has demonstrated a beneficial effect on DR due to the high content of polyphenol, an antioxidant compound that inhibits angiogenesis, prevents inflammation, and facilitates vasorelaxation, all of which results in increased blood flow in the retina [100]. It also lowers plasma glucose levels by improving insulin sensitivity [101]. Such protective associations have been reported in a cross-sectional study and recently in a prospective study, but further longitudinal studies are required to confirm the protective association. The effect of common beverages such as milk and coffee are limited, with only one and two studies, respectively [21,26,59]. The routine diet is significantly composed of the above-listed dietary factors; thus, there is a need for large-scale longitudinal studies to understand their influence on the incidence and progression of DR.

The existing guidelines from the American Diabetic Association’s (ADA) 2022 Diabetes Standard of Care support our findings, such as the benefits of the Mediterranean diet and the consumption of fruits, vegetables, and dietary fiber in cases of diabetes [102]. The ADA also recommends an increased intake of fish containing omega-3 fatty acids, which are also seen to be effective in DR prevention in our review. The evidence regarding the benefits of antioxidant supplements is insufficient in both the research of the ADA and our review. The ADA recommends limited sodium and carbohydrate consumption; however, we found no conclusive evidence to suggest detrimental effects of increased sodium and carbohydrate. Likewise, ADA recommends PUFAs and MUFAs intake as a replacement for saturated fat. It supports modest alcohol consumption, but our study results remain inconclusive regarding the effect of MUFA / PUFA and average alcohol intake on DR [102]. The findings from our review study are intended to complement and be considered simultaneously with the existing dietary guidelines in the overall management of diabetes.

### 4.4. Strengths and Limitations

The systematic review has several strengths as a method. Firstly, most studies in our review had good methodological and study qualities. Secondly, only dietary intake exposure and DRs outcome within human subjects were evaluated, excluding experimental animal and biomarker studies. This allowed us to translate results into nutritional recommendations for patients. Thirdly, studies conducted on diverse populations were included, thus providing more generalized results. However, our study also has some limitations, which may cause inconclusive outcomes between dietary intake and DR. First, FFQs were mostly used in dietary assessment and were administered only once, at the study baseline. Its major limitation is inaccurate assessment due to recall bias and subjectivity across individuals. Thus, combining methods such as the FFQ with dietary records (or 24 h dietary recall) or the FFQ with biomarker levels would provide more accurate estimates of nutritional intakes than a single assessment [103]. Second, most studies were cross-sectional, limiting the establishment of a causal association of dietary factors with DR; thus, there is a need for more longitudinal studies. Third, most studies have evaluated a single dietary component or nutrient rather than a dietary pattern that examines the effects of the overall diet. Instead of focusing on a single nutrient, broader dietary patterns, including beverages, would reflect real-world food consumption habits, which would be more predictive of disease risk and help to translate into more precise dietary guidelines [104]. Fourth, only one study evaluated the influence of dietary intake on DME; thus, there is a need for future studies in order to establish a better knowledge of the mechanisms of diet on DME, which may differ from DR. Fifth, many studies did not differentiate the effect of dietary intake on type 1 and type 2 diabetes or other types of diabetes such as gestational or autoimmune which is needed as etiology, pathophysiology, epidemiology, and disease management are not similar in a different type of diabetes. Lastly, methods assessing dietary intake exposure and DR outcomes are heterogeneous, thus affecting comparability. For example, the number of DR cases in studies examined by two-field or non-mydriatic fundus photographs may be underestimated compared to studies that used stereoscopic 7-field fundus photographs (the standard reference for DR detection as defined by the ETDRS) [105]. Therefore, further studies should be conducted on all different types of diabetes.

## 5. Conclusions

DR affects one-third of individuals with diabetes, and multiple studies depict the association between dietary intake and diabetic eye changes. While we do not fully understand the underlying mechanism that results in or worsens DR and/or DME in people with various dietary intakes, they are likely to influence glycemic management and cardiovascular risk factors. Nonetheless, diabetic patients at risk of developing DR may benefit from nutritional recommendations, as elucidated by the studies described.

## Figures and Tables

**Figure 1 nutrients-14-05021-f001:**
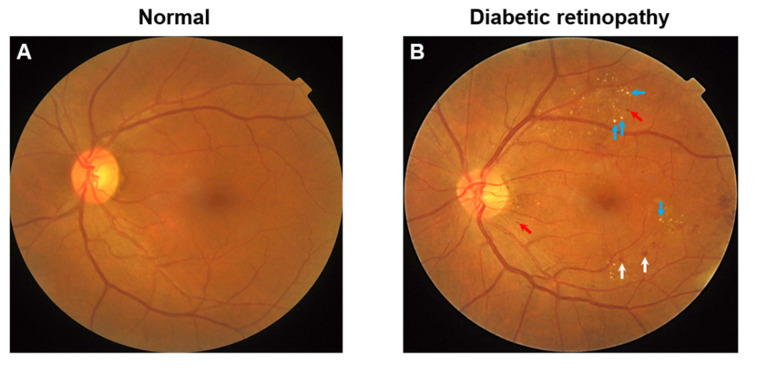
(**A**) Color fundus photograph of a diabetic individual without retinopathy. (**B**) Color fundus photograph of a diabetic individual with signs of moderate non-proliferative diabetic retinopathy. Notably, features include microaneurysms (red arrows), dot-and-blot hemorrhages (white arrows), and hard exudates (blue arrows, HE).

**Figure 2 nutrients-14-05021-f002:**
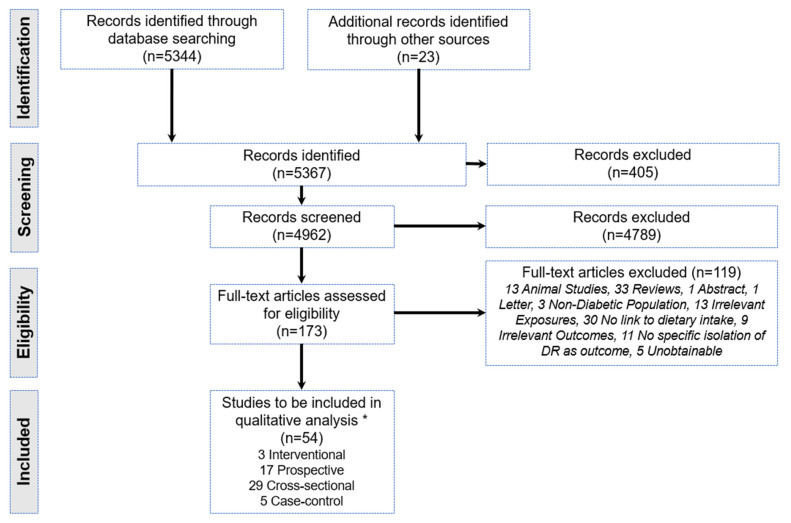
PRISMA flow diagram for the systematic review detailing the database searches, the number of abstracts screened, and the full texts retrieved. * Some studies analyzed >1 dietary component.

**Figure 3 nutrients-14-05021-f003:**
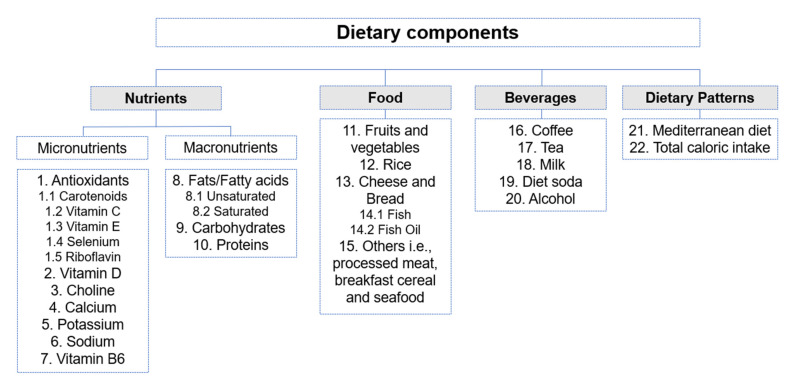
An overview of dietary components based on the studies included in the systematic review. The number assigned to the dietary component corresponds to the results section for easy referencing.

**Figure 4 nutrients-14-05021-f004:**
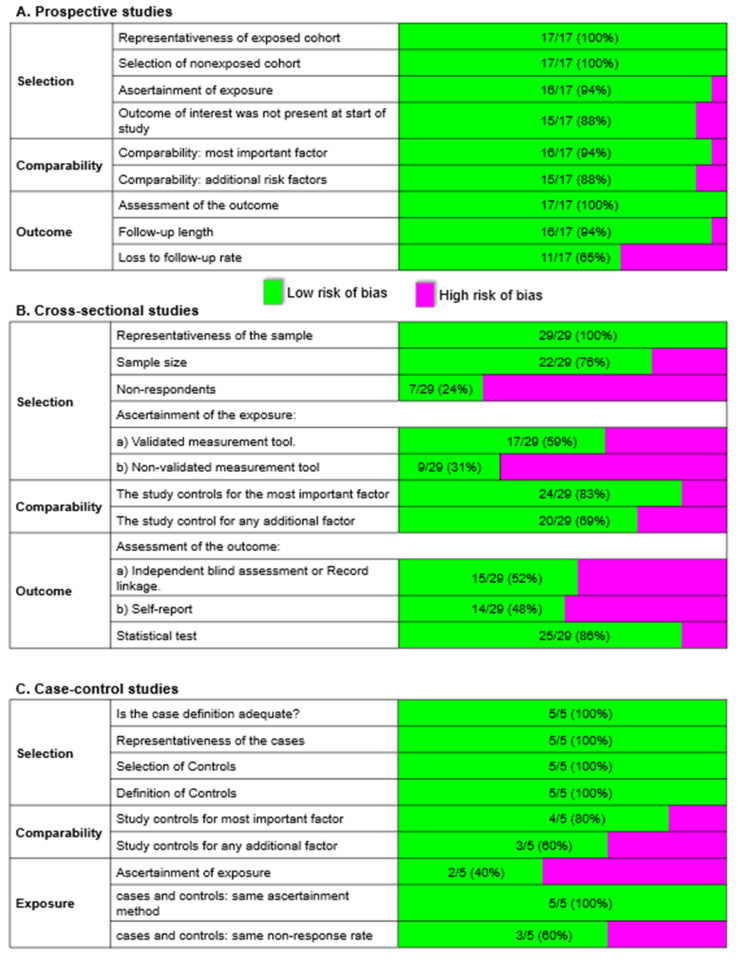
Risk of bias by the domain (in bold) and specific regions in (**A**) 17 prospective, (**B**) 29 cross-sectional, and (**C**) 5 case–control studies using the Newcastle–Ottawa Scale. Numbers on the green bar represent the number of studies with a low risk of bias over the number of studies assessed.

**Table 1 nutrients-14-05021-t001:** Characteristics of studies (n = 54).

Study, Year Sample Size	Diabetes Type	Age	Dietary Factor	Diet Evaluation	DR Outcome	DR Evaluation	Classification of DR	Quality Score
3 Interventional studies									
Houtsmuller et al., 1979 n = 96	Any diabetes	Not stated	Saturated fat diet vs. unsaturated fat diet	NA	Incidence and progression	Fundus photograph	None, NPDR, PDR, PRP	High bias
Howard-Williams et al., 1985 n = 149	Any diabetes	<66	Saturated fat diet vs. unsaturated fat diet	NA	Incidence	Ophthalmologist examination	None, retinopathy	High bias
Diaz-Lopez et al., 2015 n = 3614	T2DM	55–80	Mediterranean diet	NA	Incidence	Ophthalmologist examination	None, NPDR, PDR	Moderate bias
17 Prospective studies									
Horikawa et al., 2021 T2DM: 912	T2DM	65–85	Sodium	Validated food frequency questionnaire	Incidence	Ophthalmologist examination	Japanese Diabetes Complication Study Method	10
Park et al., 2021 DR: 731 no DR: 1336	T2DM	DR: 53.1 (9.7) no DR: 55.6 (9.7)	Glutamic acid and aspartic acid	3-day food record with computer-aided nutritional analysis	Incidence	Fundus photograph, OCT	ETDRS	10
Horikawa et al., 2017 n = 936	T2DM	40–70	Carbohydrates	Validated food frequency questionnaire	Incidence and progression	Ophthalmologist examination	International Classification System	10
Horikawa et al., 2014 n = 978	T2DM	40–70	Sodium	Validated food frequency questionnaire	Incidence and progression	Ophthalmologist examination	International Classification System	10
Tanaka et al., 2013 n = 978	T2DM	40–70	Vitamin C, Vitamin E, carotenoids, fruits, and vegetables,	Validated food frequency questionnaire + 24 h dietary recall	Incidence	Ophthalmologist examination	International Classification System	10
Hainsworth et al., 2019 PDR: 379 no PDR: 1061	T1DM	PDR: 26 (21–32) no PDR: 27 (22–32)	Alcohol beverage	Simple background questionnaire	Incidence and progression	Standardized stereoscopic seven-field fundus photographs	ETDRS	9
Horikawa et al., 2019 n = 978	T2DM	40–70	Vitamin B6	Validated food frequency questionnaire	Incidence	Mydriatic indirect ophthalmoscopic examination and slit lamp biomicroscopic fundus examination, with supplementation of fundus photography and fluorescein angiography	International Clinical Diabetic Retinopathy, DME Severity Scale	9
Sala-Vila et al., 2016 n = 3482	T2DM	55–80	Long-chain omega-3 polyunsaturated fatty acids and oily Fish	Validated food frequency questionnaire	Incidence	Clinical and hospital records	None, NPDR, PDR	9
Lee et al., 2010 n = 1239	T2DM	55–81	Alcohol	Self-report in a general questionnaire	Progression	Fundus photograph	Modified ETDRS	9
Roy et al., 2010 n = 469	T1DM	Men: 26.7 (10.7) Women: 27.8 (10.8)	MUFA, PUFA, oleic acid, protein, dietary fiber, carbohydrates, sodium, total calories, alcohol	Validated food frequency questionnaire	Incidence and progression	Fundus photograph	Modified ETDRS	9
Moss et al., 1993 Young: 439 Older: 478	Any diabetes	21–94	Alcohol	Self-report in a general questionnaire	Incidence and progression	Fundus photograph	Modified ETDRS	9
Gupta et al., 2020 Abstainers: 563 Consumers: 93	Not stated	Abstainers: 58.88 (9.45) Consumers: 58.41 (8.09)	Alcohol	Questionnaire on alcohol consumption	Incidence and progression	Fundus photograph	ETDRS, Airlie House Classification	8
Cundiff et al., 2005 n = 1412	T1DM	13–39	MUFA, PUFA, carbohydrates, protein, dietary fiber, sodium, alcohol, high calories	Dietary history interview	Progression	Fundus photograph	Modified ETDRS	8
Young et al., 1984 n = 296	Any diabetes	20–59	Alcohol	Self-report in a general questionnaire	Incidence	Direct ophthalmoscopy	Modified ETDRS	8
Ghaemi et al., 2021 T1DM with MD: 1669 T1DM without MD: 180 T2DM with MD: 15886 T2DM without MD: 4452	T1DM or T2DM	T1DM with MD: 50.63 (20.11) T1DM without MD: 51.40 (16.20) T2DM with MD: 59.78 11.00)	Mediterranean diet	14 item questionnaire	Incidence	Records from the National Program for Prevention and Control of Diabetes of Iran database	International Classification of Diseases, 10th Revision: E10.3, E11.3, E12.3, E13.3, and E14.3	7
Kadri et al., 2021 DR: 106 no DR: 155	T2DM	57.73 (11.29)	Alcohol, antioxidants, milk, tea, coffee, fruits, meat, fish, egg, chapathi, rice, total Calories	24 h dietary recall	Incidence and progression	Dilated fundus examination using slit-lamp biomicroscopy (90D), indirect ophthalmoscopy, fundus photography	Not stated	7
Yan et al., 2019 n = 8122	Not stated	57.2 (5.2)	Meat, dairy products, wholemeal bread, breakfast cereal, vegetables, fruit, and fruit juice	Self-administered questionnaire	Incidence and progression	Retinal photocoagulation from the Medicare Benefits Schedule data (note: used as a proxy for DR progression)	Not stated	6
29 Cross-Sectional Studies									
Fenwick et al., 2015 n = 395	T2DM	>18	Alcohol	Validated food frequency questionnaire	Prevalence	Non-dilated fundus photography	ETDRS	10
Ganesan et al., 2012 n = 1261	Any diabetes	>40	Dietary fiber	Validated fiber questionnaire	Prevalence	Dilated fundus photograph	Modified ETDRS	10
Beulens et al., 2008 n = 1857	T1DM	15–60	Alcohol	Self-report in a general questionnaire	Prevalence	Dilated fundus photograph	None, Background, Proliferative	10
Lee et al., 2022 DR: 270 no DR: 1080	T2DM	DR: 59.9(0.8) no DR: 58.6(0.4)	Coffee	Validated food frequency questionnaire	Prevalence	Fundus photograph	ETDRS, modified Airlie House Classification	9
Liu et al., 2021 DR: 378 no DR: 894	Not stated	>40	Choline	24 h dietary recall	Prevalence	Fundus photograph	Not stated	9
Millen et al., 2016 n = 1305	Any diabetes	45–65	Vitamin D, fish, milk	Validated food frequency questionnaire	Prevalence	Fundus photograph	Modified Airlie House Classification	9
Sahli et al., 2016 n = 1430	Any diabetes	45–65	Carotenoids (lutein)	Validated food frequency questionnaire	Prevalence	Non-dilated fundus photograph	ETDRS	9
Mayer-Davis et al., 1998 n = 387	T2DM	20–74	Vitamin C, Vitamin E, beta-carotene	24 h dietary recall	Prevalence	Dilated fundus photograph	Modified Airlie House Classification	9
Moss et al., 1992 Young: 891 Older: 987	Any diabetes	2–96	Alcohol	Self-report in a general questionnaire	Prevalence	Fundus photograph	Modified Airlie House Classification	9
Chen et al., 2022 DR: 696 no DR: 4515	Not stated	DR: 62.43 (11.79) no DR: 58.961 (12.421)	Calcium and potassium	24 h dietary recall	Prevalence	Fundus photograph	ETDRS	8
She et al., 2020 DR: 119 No DR: 336	T2DM	DR: 63.2 (8.5) no DR: 65.4 (8.8)	Antioxidants	3-day food record	Prevalence	Fundus photograph	ETDRS	8
Chua et al., 2018 n = 357	T2DM	58 (52–62)	Fish	Validated food frequency questionnaire	Prevalence	Two-field digital retinal photographs	ETDRS, Airlie House Classification	8
Fenwick et al., 2018 n = 609	T1DM or T2DM	64.6(11.6)	Diet soft drink	Validated food frequency questionnaire	Prevalence	Two-field (macula and optic disc) dilated fundus photos were captured using a non-mydriatic retinal camera (fundus photography)	ETDRS for DR and the American Academy of Ophthalmology Scale for DME	8
Granado-Casas et al., 2018 DR: 103 no DR: 140	T1DM	DR: 46.2(10.8) no DR: 42.1(10.3)	Fat	Validated food frequency questionnaire	Prevalence	Ophthalmologist examination	International Clinical Classification System for diabetic retinopathy	8
Thapa et al., 2018 DM: 1692 no DM: 168	Not stated	DM: 69.8 (7.4) no DM: 67.9 (6.7)	Alcohol	Simple background questionnaire	Prevalence	Dilated fundus examination by a retina specialist	ETDRS	8
Sasaki et al., 2015 n = 379	Any diabetes	>18	Vitamin C, Vitamin E, beta-carotene, MUFA, PUFA, carbohydrates, protein	Validated food frequency questionnaire	Prevalence	Fundus photograph	Modified ETDRS	8
Kumari et al., 2014 n = 353	Any diabetes	21–95	Coffee	Questionnaire on coffee consumption	Prevalence	Dilated fundus photograph	Modified Airlie House Classification	8
Mahoney et al., 2014 n = 155	Any diabetes	>40	Fruits and vegetables	Validated food frequency questionnaire	Prevalence	Non-dilated fundus photograph	ETDRS	8
Harjutsalo et al., 2013 n = 3608	T1DM	Median age: 37.4 (28.9–46.8)	Alcohol	Self-report in a general questionnaire	Prevalence	History of laser photocoagulation	Severe DR vs. None	8
Millen et al., 2004 n = 1353	Any diabetes	45–65	Vitamin C and Vitamin E	Validated food frequency questionnaire	Prevalence	Non-dilated fundus photograph	Modified Airlie House Classification	8
Xu et al., 2020 DM: 614 no DM: 4667	Not stated	DM: 68.03(6.49) no DM: 67.88(6.64)	Tea	Questionnaire on tea consumption	Prevalence	Fundus photograph	ETDRS	7
Engelen et al., 2014 n = 1880	T1DM	15–60	Sodium	Estimated from urinary sodium excretion	Prevalence	Fundus photograph	None, NPDR, PDR	7
Shalini et al., 2021 DR: 194 no DR: 150 Control: 151	T2DM	DR: 55.0(0.6) no DR: 56.0(0.9) Control: 54.0(0.9)	Carotenoids	Validated raw food-based food frequency questionnaire with HPLC of plasma carotenoids	Prevalence	Fundus examination by indirect ophthalmoscopy, slit-lamp biomicroscopy, fundus fluorescein angiography	ETDRS	6
Alsbirk et al., 2021 T1DM: 50 T2DM: 460	T1DM or T2DM	T1DM: 44.5 (13–87) T2DM: 66 (27–92)	Fish food, PUFAs supplements	Questionnaire of self-reported dietary history	Prevalence	Fundus photograph	International Clinical Diabetic Retinopathy, DME Severity Scale	6
Mirghani et al., 2021 DR: 66 no DR: 134	Not stated	50.74(13.51)	Diet sugar-free carbonated soda beverage	Validated food frequency questionnaire	Prevalence	Fundus examination	Not stated	5
Kawasaki et al., 2018 NPDR: 83 no NPDR: 280	T1DM or T2DM	NPDR: 58.9 no NPDR: 55.6	Alcohol	Simple background questionnaire	Prevalence	Fundus findings from clinic and hospital records	International Clinical Diabetic Retinopathy	5
Lugo-Radillo et al., 2013 n = 88	Any diabetes	No DR: 58.50 (1.11) DR: 56.82 (1.65)	Fruits and vegetables	Oral questionnaire on fruit and vegetable consumption	Prevalence	Ophthalmologist examination	International Classification System	5
Roy et al., 1989 n = 34	Any diabetes	DR: 37.9 (12) No DR: 37.7 (9)	MUFA, PUFA, carbohydrates, protein, dietary fiber	3-day food record	Prevalence	Fundus photography	Modified Airlie House Classification	5
Acan et al., 2018 DME: 63 no DME: 350	T1DM or T2DM	DME: 58.86 (11.27) no DME: 56.03 (11.95)	Alcohol	Simple background questionnaire	Prevalence	Dilated fundoscopy by ophthalmologists, central macular thickness analysis with OCT	ETDRS, OCT central macular thickness ≥ 250 μm	3
5 Case–control Studies									
Alcubierre et al., 2016 Case: 146 Control:148	T2DM	40–75	MUFA, PUFA, oleic acid, carbohydrates, protein, dietary fiber	Validated food frequency questionnaire	Prevalence	Ophthalmologist examination	International Classification System	10
Zhang et al., 2019 DM with DR: 43 DM without DR: 43 Controls: 40	T2DM	DM with DR: 59 (49–66) DM without DR: 53 (44–65) Controls: 54(47–67)	Vitamin A	Validated food frequency questionnaire with HPLC of plasma retinol	Prevalence	Not stated	Not stated	8
Alcubierre et al., 2015 Case: 139 Control:144	T2DM	No DR: 58.1 (10.3) DR: 60.3 (8.9)	Vitamin D, calcium	Validated food frequency questionnaire	Prevalence	Ophthalmologist examination	International Classification System	8
Ma et al., 2014 Case:100 Control:100	T2DM	>18	Green tea	Questionnaire on tea consumption	Prevalence	Fundus photograph	ETDRS	8
Giuffre et al., 2004 Case:45 Control:87	Any diabetes	>40	Alcohol	Self-report in a general questionnaire	Prevalence	Direct ophthalmoscopy and fundus photograph	ETDRS	7

DR—Diabetic retinopathy, DME—Diabetic macular edema, ETDRS—Early treatment diabetic retinopathy study, HPLC—High-performance liquid chromatography, MD—Mediterranean diet, MUFA—Monounsaturated fatty acid, NPDR—Non-proliferative diabetic retinopathy, OCT—Optical coherence tomography, PDR—Proliferative diabetic retinopathy, PRP—Pan retinal photocoagulation, PUFA—polyunsaturated fatty acid, DM—Diabetes Mellitus.

**Table 2 nutrients-14-05021-t002:** Dietary intake of micronutrients and diabetic retinopathy.

Study, Year Study Design Sample Size (n)	Quality Score	Dietary Factor and Its Association with DR	Adjustment/Matched	Statistical Method Analysis	Key Findings
Antioxidants					
Carotenoids					
Tanaka et al., 2013 Prospective n = 978	10	Carotenoids Protective	Sex, age, BMI, HbA1c, diabetes duration, insulin treatment, oral hypoglycaemic agents without insulin treatment, systolic blood pressure, LDL and HDL cholesterol, triglycerides, physical activity alcohol, smoking, total energy intake, proportions of dietary protein, fat, carbohydrate, saturated fatty acids, omega-6 PUFA and omega-3 PUFA and sodium	Multivariate Cox regression	Highest intake Q4 vs. lowest Intake Q1, HR: 0.52 (0.33–0.81) *p* < 0.01
Sahli et al., 2016 Cross-sectional n = 1430	9	Lutein carotenoids NS	Diabetes duration, HbA1c, blood pressure, race, total energy consumption, and study center	Multivariable logistic regression	Intake Q4 vs. Q1, OR: 0.89 (0.31–2.50), *p* = 0.72
Mayer-Davis et al., 1998 Cross-sectional n = 387	9	Beta-Carotene NS	Age, gender, ethnicity, diabetes duration, HbA1c, hypertension, caloric intake, and insulin use	Multivariable logistic regression	No significant associations with DR (data not shown)
Zhang et al., 2019 Case–control Type2 DM-86 control-40	8	Retinol carotenoids Protective	Age, sex, smoking, BMI and alcohol consumption	Logistic regression	Intake of retinol (100 μg/day) on DR (OR: 0.88, 95%CI, 0.79–0.98, *p* = 0.025)
Sasaki et al., 2015 Cross-sectional n = 379	8	Beta-carotene NS	Intake of energy	Data not shown	No significant associations with DR (data not shown)
Shalini et al., 2021 Cross-sectional n = 495	7	Carotenoids Protective	Nil	One-way analysis of variance F test with a post hoc test of least significant difference	The plasma concentration of carotenoids was significantly lower in the DR group compared to no DR patients and healthy controls (*p* < 0.001)
Vitamin C					
Tanaka et al., 2013 Prospective n = 978	10	Vitamin C Protective	Sex, age, BMI, HbA1c, diabetes duration, insulin treatment, oral hypoglycaemic agents without insulin treatment, systolic blood pressure, LDL and HDL cholesterol, triglycerides, physical activity alcohol, smoking, total energy intake, proportions of dietary protein, fat, carbohydrate, saturated fatty acids, omega-6 PUFA and omega-3 PUFA and sodium	Multivariate Cox regression	Intake Q4 vs. Q1, HR: 0.61 (0.39–0.96), *p* = 0.03
Mayer-Davis et al., 1998 Cross-sectional N = 387	9	Vitamin C Risk	Age, gender, ethnicity, diabetes duration, HbA1c, hypertension, caloric intake, and insulin use	Multivariable logistic regression	Intake 9th decile vs. 1st quintile, OR: 2.21, (*p* = 0.01)
She et al., 2020 Cross-sectional n = 455	8	Vitamin C NS	Sex, race, insulin use, HbA1c, hypertension, exercise	Binomial logistic regression multivariate analysis	No significant association with DR (*p* = 0.413)
Sasaki et al., 2015 Cross-sectional n = 379	8	Vitamin C NS	Intake of energy	Data notshown	No significant association with DR (data not shown)
Millen et al., 2004 Cross-sectional n = 1353	8	Vitamin C NS	Race, BMI, diabetes duration, serum glucose, total energy intake, hypertension, waist–hip ratio, smoking, alcohol, drinking status, plasma cholesterol, hematocrit value, prevalent coronary heart disease, plasma triacylglycerol, diabetes treatment group, and oral hypoglycaemic treatment or insulin treatment	Multivariable logistic regression	Intake Q4 vs. Q1, OR: 1.4 (0.8–2.4), *p* = 0.19
Vitamin E					
Tanaka et al., 2013 Prospective n = 978	10	Vitamin E NS	Sex, age, BMI, HbA1c, diabetes duration, insulin treatment, oral hypoglycaemic agents without insulin treatment, systolic blood pressure, LDL and HDL cholesterol, triglycerides, physical activity alcohol, smoking, total energy intake, proportions of dietary protein, fat, carbohydrate, saturated fatty acids, omega-6 PUFA and omega-3 PUFA and sodium	Multivariate Cox regression	Intake Q4 vs. Q1, HR: 0.84 (0.51–1.40), *p* = 0.51
Mayer-Davis et al., 1998 Cross-sectional N = 387	9	Vitamin E Risk (in non-insulin taking subjects)	Age, gender, ethnicity, diabetes duration, HbA1c, hypertension, caloric intake, and insulin use	Multivariable logistic regression	No association found in insulin subjects and in non-insulin taking subjects: Intake 10th decile vs. 1st quintile, OR: 3.79, (*p* < 0.02)
She et al., 2020 Cross-sectional n = 455	8	Vitamin E Protective	Sex, race, insulin use, HbA1c, hypertension, exercise	Binomial logistic regression multivariate analysis	Intake in DR vs. No DR (OR: 0.97, 95%CI: 0.95–1.00, *p* = 0.036)
Granado-Casas et al., 2018 Cross-sectional n = 243	8	Vitamin E Protective	Age, sex, educational level, smoking, physical activity, BMI, dyslipidemia, hypertension, diabetes duration, HbA1c	Multivariable conditional logistic regression models	Intake of Vitamin E on DR (OR: 0.85 [0.77–0.95], *p* = 0.006)
Sasaki et al., 2015 Cross-sectional n = 379	8	Vitamin E NS	Intake of energy	Data notshown	No significant associations with DR (data not shown)
Millen et al., 2004 Cross-sectional n = 1353	8	Vitamin E NS	Race, BMI, diabetes duration, serum glucose, total energy intake, hypertension, waist–hip ratio, smoking, alcohol, drinking status, plasma cholesterol, hematocrit value, prevalent coronary heart disease, plasma triacylglycerol, diabetes treatment group, and oral hypoglycaemic treatment or insulin treatment	Multivariable logistic regression	Intake Q4 vs. Q1, OR: 1.4 (0.8–2.3), *p* = 0.76
Selenium					
She et al., 2020 Cross-sectional n = 455	8	Selenium Protective	Sex, race, insulin use, HbA1c, hypertension, exercise	Binomial logistic regression multivariate analysis	Intake in DR vs. No DR (OR: 0.98, 95%CI: 0.96–1.00, *p* = 0.017)
Riboflavin					
She et al., 2020 Cross-sectional n = 455	8	Riboflavin NS	Sex, race, insulin use, HbA1c, hypertension, exercise	Binomial logistic regression multivariate analysis	No significant association with DR (*p* > 0.05)
Vitamin D					
Millen et al., 2016 Cross-sectional n = 1305	9	Vitamin D NS	Race, duration of diabetes, HbA1c and, hypertension	Multivariable logistic regression	Intake Q4 vs. Q1, OR: 1.20 (0.76–1.89), *p* trend = 0.740
Alcubierre et al., 2015 Case–control Case:139 Ctrl:144	8	Vitamin D NS	NIL	Chi-squared	No significant associations with DR (*p* = 0.93)
Choline					
Liu et al., 2021 Cross-sectional n = 1272	9	Choline Risk in female	Age, race, diabetes duration, glycaemic control, CVD, CKD * results analyzed in individual sex groups	Multivariable logistic regression	High intake vs. low intake (OR: 2.14, 95%CI: 1.38–3.31; *p* = 0.001)
Calcium					
Chen et al., 2022 Cross-sectional n = 5321	9	Calcium Protective	Age, sex, race, smoking, serum glucose, serum laboratory data, hemoglobin	Multivariable logistic regression	High intake vs. low intake OR: 0.70, 95%CI: 0.54–0.90, *p* = 0.05)
Alcubierre et al., 2015 Case–control Case:139 Ctrl:144	8	Calcium NS	NIL	Chi-squared	No significant associations with DR (*p* = 0.65)
Potassium					
Tanaka et al., 2013 Prospective n = 978	10	Potassium NS	Sex, age, BMI, HbA1c, diabetes duration, insulin treatment, oral hypoglycaemic agents without insulin treatment, systolic blood pressure, LDL and HDL cholesterol, triglycerides, physical activity alcohol, smoking, total energy intake, proportions of dietary protein, fat, carbohydrate, saturated fatty acids, omega-6 PUFA and omega-3 PUFA and sodium	Multivariate Cox regression	No significant association with DR (*p* > 0.05)
Chen et al., 2022 Cross-sectional n = 5321	9	Potassium Protective	Age, sex, race, smoking serum glucose, serum laboratory data, hemoglobin	Multivariable logistic regression	High intake vs. low intake OR: 0.761, 95%CI: 0.59–0.97, *p* = 0.029
Sodium					
Horikawa et al., 2021 Prospective n = 912	10	Sodium Risk (under low vegetable consumption)	Age, sex, BMI, HbA1c, diabetes duration, LDL cholesterol, HDL cholesterol, log-transformed triglycerides, insulin treatment, smoking, alcohol, energy intake, physical activity, systolic blood pressure, angiotensin II receptor blocker, angiotensin-converting enzyme inhibitor, calcium channel blocker	Multivariate Cox regression analyses	Intake for 2nd, 3rd, and 4th quartile vs. 1st quartile, HRs were 0.87 [95%CI, 0.31–2.41], 2.61 [1.00–6.83], and 3.70 [1.37–10.02], respectively *p* = 0.010.
Horikawa et al., 2014 Prospective n = 978	10	Sodium NS	Sex, age, BMI, HbA1c, duration of diabetes, LDL cholesterol, HDL cholesterol, log-transformed triglycerides, insulin treatment, lipid-lowering agents, smoking, alcohol intake, energy intake, sodium intake, and physical activity	Multivariate Cox regression	Intake Q4 vs. Q1, HR: 1.10 (0.75–1.61), *p* = 0.55
Roy et al., 2010 Prospective n = 469	10	Sodium Risk (ForDME) NS for DR	Age, sex, HbA1c, hypertension, total caloric intake, protein intake, oleic acid intake, physical exercise, and oleic acid intake	Multivariable logistic regression	Intake Q4 vs. Q1, OR: 1.43 (1.10–1.86), *p* = 0.008 for DME. No significant associations with DR
Cundiff et al., 2005 Prospective n = 1412	8	Sodium NS	Intake of energy	Spearman correlation	Sodium in mg/kcal against DR progression rate, r = 0.02 (*p* = 0.47)
Engelen et al., 2014 Cross-sectional n = 1880	7	Sodium NS	Sex, age, smoking, BMI, urinary potassium excretion, sat fat intake, protein intake antihypertensive medication, total energy intake, physical activity, fiber intake, and alcohol intake	Multivariable logistic regression	Per 1g/day increase in dietary salt intake against NPDR OR: 1.00, (0.96–1.04, *p* = 0.84. PDR OR: 1.02 (0.95–1.08), *p* = 0.65
Vitamin B6					
Horikawa et al., 2019 Prospective n = 978	9	Vitamin B6 Protective	Age, sex, BMI, HbA1c, diabetes duration, systolic blood pressure, LDL cholesterol, HDL cholesterol, triglycerides, insulin treatment, oral hypoglycemic agents, antihypertensive agents, lipid-lowering agents, urine albumin creatinine ratio, estimated glomerular filtration rate, alcohol, smoking, energy intake, physical activity, retinol, vitamin B1, vitamin B2, vitamin B9, vitamin B12	Multivariate Cox regression analyses	Intake Q4 vs. Q1 HR: 0.50, 95%CI: 0.30–0.85, *p* = 0.010)

BMI—Body mass index, CI–Confidence interval, CVD—Cardiovascular disease, CKD—Chronic kidney disease, CI—Confidence interval, DR—Diabetic retinopathy, DME—diabetic macular edema, DM—Diabetes mellitus, HDL—High-density lipoprotein, HR—Hazard ratio, HbA1c—glycated hemoglobin, LDL—Low-density lipoprotein, NS—Not significant, NPDR—Non-proliferative diabetic retinopathy, OR—Odds ratio, PUFA—Polyunsaturated fatty acid, PDR—Proliferative diabetic retinopathy.

**Table 3 nutrients-14-05021-t003:** Dietary intake of macronutrients and diabetic retinopathy.

Study, Year Study Design Sample Size (n)	Quality Score	Dietary Factor and Its Association with DR	Adjustment/Matched	Statistical Methods Analysis	Key Findings
Dietary Fats/lipids					
Monounsaturated Fatty Acids (MUFA)					
Alcubierre et al.,2016 Case–control Case:146 Ctrl:148	10	MUFA Protective	Sex, age, diabetes duration, energy intake, systolic blood pressure, physical activity, waist circumference, HDL cholesterol, educational level and diabetes treatment	Multivariable logistic regression	High MUFA consumption vs. low MUFA consumption, OR: 0.42 (0.18–0.97), *p* = 0.034
Sasaki et al., 2015 Cross-sectional n = 379	10	MUFA NS	Sex, Age, HbA1c, duration of diabetes, and mean arterial pressure	Multivariable logistic regression models	Per 10 energy-adjusted g/d increase, OR: 1.19 (0.74–1.92)
Roy et al., 2010 Prospective n = 469	9	MUFA NS	Total fat, total caloric intake, oleic acid, linoleic acid, fiber, protein, sat fat, cholesterol and sodium intakes	Multivariable logistic regression	No significant associations with DR (data not shown)
Granado-Casas et al., 2018 Cross-sectional n = 243	8	MUFA Protective	Age, sex, educational level, smoking, center, physical activity, BMI, dyslipidemia hypertension, diabetes duration, HbA1c	Multivariable conditional logistic regression models	MUFA intake against frequency of DR (OR: 0.95, 95%CI: 0.92–0.99, *p* = 0.012)
Cundiff et al., 2005 Prospective n = 1412	8	MUFA Risk	Intake of energy	Spearmancorrelation	MUFA in %/kcal against DR progression rate, r = 0.12 (*p* = 0.001)
Roy et al., 1989 Cross-sectional n = 34	5	MUFA NS	Intake of energy	*t* test	No significant associations with DR (data not shown)
Oleic acid					
Alcubierre et al., 2016 Case–control Case:146 Ctrl:148	10	Oleic acid Protective	Sex, age, diabetes duration, energy intake, systolic blood pressure, physical activity, waist circumference, HDL cholesterol, educational level and diabetes treatment	Multivariable logistic regression	Highest intake tertile (T3) vs. lowest intake tertile (T1), OR: 0.37 (0.16–0.85), *p* = 0.017
Roy et al., 2010 Prospective n = 469	9	Oleic acid NS	Total fat, total caloric intake, oleic acid, linoleic acid, fiber, protein, sat fat, cholesterol and sodium intake	Multivariable logistic regression	No significant associations with DR (data not reported)
Granado-Casas et al., 2018 Cross-sectional n = 243	8	Oleic acid Protective	Age, sex, educational level, smoking, center, physical activity, BMI, dyslipidemia hypertension, diabetes duration, HbA1c	Multivariable conditional logistic regression models	Oleic acid intake against DR (OR: 0.95, CI: 0.92–0.99, *p* = 0.012)
Polyunsaturated Fatty Acids (PUFA)					
Alcubierre et al., 2016 Case–control Case:146 Ctrl:148	10	PUFA NS	Sex, age, diabetes duration, energy intake, systolic blood pressure, physical activity, waist circumference, HDL cholesterol, educational level and diabetes treatment	Multivariable logistic regression	High PUFA consumption vs. low MUFA consumption, OR: 0.99 (0.69–1.41)
Sasaki et al., 2015 Cross-sectional n = 379	10	PUFA Protective forwell controlleddiabetics	Sex, age, HbA1c, duration of diabetes, and mean arterial pressure	Multivariablelogistic regression models	All subjects: Per 10 energy-adjusted g/d increase, OR: 0.67 (0.37–1.20) Well-controlled diabetics: Per 10 energy adjusted g/d increase, OR: 0.18 (0.06–0.59)
Sala-Vila et al., 2016 Prospective n = 3482	9	PUFA (long-chain omega-3 fatty acid) Protective	Age, sex, BMI, intervention group, duration of diabetes, insulin treatment, oral hypoglycemic treatment, smoking, hypertension, systolic blood pressure, physical activity, and adherence to the Mediterranean diet	Cox proportional hazard model	>500 mg/d vs. <500 mg/d, HR: 0.52 (0.31–0.88) *p* = 0.001
Roy et al., 2010 Prospective n = 469	9	PUFA NS	Total fat, total caloric intake, oleic acid, linoleic acid, fiber, protein, sat fat, cholesterol and sodium intakes	Multivariable logistic regression	No significant associations with DR (data not shown)
Cundiff et al., 2005 Prospective n = 1412	8	PUFA Risk	Intake of energy	Spearman correlation	PUFA in %/kcal against DR progression rate, r = 0.09 (*p* = 0.004)
Roy et al., 1989 cross-sectional	5	PUFA NS	Intake of energy	*t* test	No significant associations with DR (data not reported)
Interventional studies					
Howard-Williams et al., 1985 Interventional n = 149	HighBias	PUFA NS	Age, sex and BMI	Participants on a modified fat diet (PUFA: saturated fat ratio, 0.3) vs. low-carb diet (PUFA: saturated fat ratio, 0.9) No difference between the two groups in all participants (n = 149) (chi-squared, *p* = 0.69) No difference between the two groups in dietary compliers (n = 58) (chi-squared, *p* = 0.13)	
Houtsmuller et al., 1979 Interventional n = 96	Highbias	Unsaturatedfats Protective	Gender	Saturated fat diet vs. unsaturated fat diet males (n = 52, 26 on each diet) *p* < 0.001 females (n = 44, 22 on each diet) *p* < 0.025	
Carbohydrates					
Horikawa et al., 2017 Prospective n = 936	10	Carbohydrates NS	Gender, age, BMI, HbA1c, diabetes duration, insulin treatment, systolic blood pressure, LDL cholesterol, HDL cholesterol, antihypertensive agents, lipids lowering drugs, energy intake, triglycerides, current smoker, alcohol consumption, and physical activity	Multivariable Cox regression models	Highest intake tertile (T3) vs. lowest intake tertile (T1), HR: 1.00 (0.72–1.38)
Alcubierre et al., 2016 Case–control Case:146 Ctrl:148	10	Carbohydrates NS	Sex, age, diabetes duration, energy intake, systolic blood pressure, physical activity, waist circumference, HDL cholesterol, educational level and diabetes treatment	Multivariable logistic regression	Highest intake tertile (T3) vs. lowest intake tertile (T1), OR: 1.18 (0.45–3.09)
Roy et al., 2010 Prospective n = 469	9	Carbohydrates NS	Total fat, total caloric intake, oleic acid, linoleic acid, fiber, protein, sat fat, cholesterol, and sodium intakes	Multivariable logistic regression	No significant associations with DR (data not shown)
Granado-Casas et al., 2018 Cross-sectional n = 243	8	Carbohydrates Risk	Age, sex, educational level smoking, center, physical activity, BMI, dyslipidemia hypertension, diabetes duration, HbA1c	Multivariable conditional logistic regression models	Intake of complex carbohydrates against DR (OR: 1.02, CI: 1.00–1.04, *p* = 0.031)
Sasaki et al., 2015 Cross-sectional n = 379	8	Carbohydrates NS	Intake of energy	Chi-squared	No significant associations with DR (data not shown)
Cundiff et al., 2005 Prospective n = 1412	8	Carbohydrates Protective	Intake of energy	Spearman correlation	Carbohydrates in %/kcal against DR progression rate, r = −0.11 (*p* < 0.001)
Roy et al., 1989 cross-sectional n = 34	5	Carbohydrates Protective	Intake of energy	*t* test	Persons without retinopathy vs. persons with retinopathy (*p* < 0.05)
Protein					
Park et al., 2021 Prospective n = 2067	10	Protein (glutamic acid and aspartic acid) NS for DR incidence, however aspartic acid protective for PDR	Age, sex, HbA1c, diabetes duration, education income, occupation, creatinine clearance, alanine aminotransferase, other comorbidities	Cox proportional hazard models	No significant association with DR incidence. Intake of aspartic acid highest tertile vs. lowest tertile for PDR (HR: 0.39, 95%CI: 0.16–0.96, *p* = 0.013)
Alcubierre et al., 2016 Case–control Case:146 Ctrl:148	10	Protein NS	Sex, age, diabetes duration, energy intake, systolic blood pressure, physical activity, waist circumference, HDL cholesterol, educational level and diabetes treatment	Multivariable logistic regression	Highest protein intake tertile (T3) vs lowest protein intake tertile (T1), OR: 1.24 (0.49–3.16)
Roy et al., 2010 Prospective n = 469	9	Protein NS	Total fat, total caloric intake, oleic acid, linoleic acid, fiber, protein, sat fat, cholesterol, and sodium intakes	Multivariable logistic regression	No significant associations with DR (data not shown)
Sasaki et al., 2015 Cross-sectional n = 379	8	Protein NS	Intake of energy	Chi-squared	No significant associations with DR (data not shown)
Cundiff et al., 2005 Prospective n = 1412	8	Protein Protective	Intake of energy	Spearman correlation	Protein in %/kcal against DR progression rate, r = −0.6 (*p* = 0.018)
Roy et al., 1989 Cross-sectional n = 34	5	Protein Risk	Intake of energy	*t* test	Persons without retinopathy vs. persons with retinopathy (*p* < 0.02)

CI—confidence interval, DME—Diabetic macular edema, DR—Diabetic retinopathy, HR—Hazard ratio, HbA1c—glycated hemoglobin, HDL—High-density lipoprotein, LDL—Low-density lipoprotein, MUFA—Monounsaturated fatty acid, NS—No significance, NPDR—Non-proliferative diabetic retinopathy, OR—Odds ratio, PDR—Proliferative diabetic retinopathy, PUFA—Polyunsaturated fatty acid.

**Table 4 nutrients-14-05021-t004:** Dietary intake of foods and diabetic retinopathy.

Study, Year Study Design Sample Size (n)	Quality score	Dietary Factor and Its Association with DR	Adjustment/Matched	Statistical Methods Analysis	Key Findings
Fruits, vegetables, and dietary fiber					
Alcubierre et al., 2016 Case–control Case:146 Ctrl:148	10	Dietary fiber NS	Sex, age, diabetes duration, energy intake, systolic blood pressure, physical activity, waist circumference, HDL cholesterol, educational level and diabetes treatment	Multivariable logistic regression	Highest fiber intake tertile (T3) vs. lowest fiber intake tertile (T1), OR: 0.76 (0.33–0.76)
Tanaka et al., 2013 Prospective n = 978	10	Fruits, vegetables, and dietary fiber Protective	Sex, age, BMI, HbA1c, diabetes duration, insulin treatment, oral hypoglycaemic agents without insulin treatment, systolic blood pressure, LDL and HDL cholesterol, triglycerides, physical activity alcohol, smoking, total energy intake, proportions of dietary protein, fat, carbohydrate, saturated fatty acids, omega-6 PUFA and omega-3 PUFA and sodium	Multivariate Cox regression	Veg and fruit intake Q4 vs. Q1, HR: 0.59 (0.37–0.92), *p* < 0.01. Fruit intake Q4 vs. Q1, HR: 0.48(0.32–0.71), *p* = 0.01. Dietary fiber intake Q4 vs. Q1, HR: 0.63 (0.38–1.03), *p* = 0.07.
Ganesan et al., 2012 Cross-sectional n = 1261	10	Dietary fiber Protective	Sex, Age, diabetes duration, blood pressure, BMI, Hba1c, serum lipids, smoking, and, socioeconomic status.	Multivariable logistic regression	Low-fiber diet vs. healthy fiber diet for any DR, OR: 1.41 (1.02–1.94), *p* = 0.039. Low-fiber diet vs. healthy fiber diet for VTDR, OR: 2.24 (1.01–5.02), *p* = 0.049.
Roy et al., 2010 Prospective n = 469	9	Dietary fiber NS	Total fat, total caloric intake, oleic acid, linoleic acid, fiber, protein, sat fat, cholesterol, and sodium intakes	Multivariable logistic regression	No significant associations with DR (Data not shown)
Cundiff et al., 2005 Prospective n = 1412	8	Dietary fiber Protective	Intake of energy	Spearman correlation	Dietary fiber in g/1000kcal against DR progression rate, r = −0.10 (*p* = 0.002)
Yan et al., 2019 Prospective n = 8122	6	Fruits, vegetables, and dietary fiber NS	Age, sex, income, educational level, BMI, hypertension, CVD, family history of diabetes, insulin treatment	Cox regression model.	No significant associations with DR (*p* < 0.05)
Roy et al., 1989 Cross-sectional n = 34	5	Dietary fiber Protective	Diabetes duration	*t* test	Persons without retinopathy vs. persons with retinopathy, (*p* < 0.01)
Rice					
Kadri et al., 2021 Prospective n = 261	8	Rice Risk	Age, sex, duration, antioxidants, pharmacological treatment, egg, fish, chapathi, rice	Multivariate regression analysis	Rice consumption yes vs. no, OR: 3.19, 95%CI: 1.17–8.69, *p* = 0.018
Cheese and wholemeal bread					
Yan et al., 2019 Prospective n = 8122	6	Cheese and wholemeal bread Protective	Age, sex, income, educational level, BMI, hypertension, CVD, family history of diabetes, insulin treatment	Cox regression model.	Cheese intake highest quartiles vs. lowest HR: 0.58, 95%CI: 0.41–0.83, *p* = 0.007 and wholemeal bread HR: 0.64, CI: 0.4–0.89, *p* = 0.04
Fish					
Sala-Vila et al., 2016 Prospective n = 3482	9	Oily fish Protective	Age, sex, BMI, intervention group, duration of diabetes, insulin treatment, oral hypoglycemic treatment, smoking, hypertension, systolic blood pressure, physical activity, and adherence to the Mediterranean diet	Cox proportional hazard model	>2 servings a week vs. <2 servings a week, HR: 0.41 (0.23–0.72), *p* = 0.002
Kadri et al., 2021 Prospective n = 261	8	Fish Protective	Age, sex, duration, antioxidants, pharmacological treatment, egg, fish, chapathi, rice	Multivariate regression analysis	Fish intake, more frequent vs. less frequent, OR: 0.42, 95%CI: 0.18–0.94, *p* < 0.05
Chua et al., 2018 Cross-sectional n = 357	8	Fish Protective	Age, sex, race, smoking diabetes duration, diabetic treatment, lipid-lowering medication use, systolic blood pressure, HbA1c, triglycerides	Ordered logistic and linear regression models	Per one serving increase in fish intake per week, OR: 0.91, 95%CI: 0.84–0.99, *p* = 0.038
Yan et al., 2019 Prospective n = 8122	6	Fish NS	Age, sex, income, educational level, BMI, hypertension, CVD, family history of diabetes, Insulin treatment	Cox regression model	No significant associations with DR (*p* = 0.22)
Alsbirk et al., 2021 Cross-sectional n = 510	6	Fish oil NS	Age, sex, diabetes type, diabetes duration, HbA1c, medication	Logistic regression	No significant association (*p* > 0.005)
Other types of food					
Yan et al., 2019 Prospective n = 8122	6	Processed meat/breakfast cereal NS	Age, sex, income, educational level, BMI, hypertension, CVD, family history of diabetes, insulin treatment	Cox regression model.	No significant associations with DR (*p* > 0.05)

BMI—Body mass index, CVD—Cardiovascular disease, DR—Diabetic retinopathy, HDL—High-density lipoprotein, HbA1c—Glycated hemoglobin, PUFA—Polyunsaturated fatty acid, VTDR—Vision-threatening diabetic retinopathy.

**Table 5 nutrients-14-05021-t005:** Dietary intake of beverages, dietary patterns, and diabetic retinopathy.

Study, Year Study Design Sample Size (n)	Quality Score	Dietary Factor and Its Association with DR	Adjustment/Matched	Statistical Methods Analysis	Key Findings
Coffee					
Lee at al, 2022 Cross-sectional n = 1350	9	Coffee Protective	Age, sex, education, income, BMI, energy intake, hypertension, dyslipidemia, diabetes duration, HbA1c, smoking, alcohol, physical activity	Multivariable logistic regression models	Consumption ≥ 2 cups coffee/day vs. none for DR (OR: 0.53, 95%CI: 0.28–0.99, p for trend = 0.025) and VTDR (OR: 0.30, 95%CI: 0.10–0.91, *p* for trend = 0.005)
Kumari et al., 2014 Cross-sectional n = 353	9	Coffee NS	Sex, age, HbA1c, smoking, BMI, creatinine, education level, diabetes duration, family history of diabetes, hypertension, stroke, ischemic heart disease, dyslipidemia, and cancer	Multivariable logistic regression	Coffee drinker vs. never/rarely, OR: 1.36 (0.69–2.69)
Tea					
Ma et al., 2014 Case–control Case:100 Ctrl:100	8	Green Tea Protective	Diabetes duration, insulin treatment, family history of diabetes, fasting blood glucose, education, BMI, systolic blood pressure, smoking, alcohol, physical and, activity	Multivariable logistic regression	Regular Chinese green tea drinker vs. non-regular Chinese green tea drinker, OR: 0.48, CI: 0.24–0.97, *p* = 0.04
Xu et al., 2020 Cross-sectional n = 5,281	7	Tea Protective	Age, sex, individual monthly income, fasting blood glucose, systolic blood pressure, occupation, educational level, smoking, alcohol	Multivariate logistic regression analyses	Tea consumers vs. non-tea consumers, OR: 0:29, 95%CI: 0.09–0.97, *p* = 0.04
Milk					
Yan et al., 2019 Prospective n = 8122	6	Milk NS	Age, sex, income, educational level, BMI, hypertension, CVD, family history of diabetes, insulin treatment	Cox regression model	No significant associations with DR (*p* = 0.74)
Diet soda					
Fenwick et al., 2018 Cross-sectional n = 609	8	Diet soft drink Risk	Age, sex, HbA1c, diabetes duration, insulin use, presence of at least one other diabetes complication, diabetes type, BMI, education antihypertensive medication, hyperlipidaemia, presence of comorbidity, smoking, alcohol energy intake, regular soft drink consumption	Multinomial logistic regression	High-consumption (>4 cans [1.5 liters]/week) vs. no consumption for proliferative DR (OR = 2.62, 95%CI = 1.14–6.06, *p* = 0.024)
Mirghani et al., 2021 Cross-sectional n = 200	6	Diet sugar-free carbonated soda beverage Risk	NIL	Multiple regression analysis	Diet soda was associated with DR (*p* = 0.043)
Alcohol					
Fenwick et al., 2015 Cross-sectional n = 395	10	Alcohol Protective	Sex, gender, poorly controlled diabetes, diabetes duration, BMI, smoking, systolic blood pressure, insulin therapy, and presence of at least one other diabetic complication	Multivariable logistic regression	Moderate vs. abstainers, OR: 0.47 (0.26–0.85), *p* = 0.013; moderate white wine vs. abstainers, OR: 0.48 (0.25–0.91), *p* = 0.024; moderate fortified wine vs. abstainers, OR: 0.15 (0.04–0.62), *p* = 0.009
Beulens et al., 2008 Cross-sectional n = 1857	10	Alcohol Protective	Sex, Age, smoking, center, smoking, diabetes duration, physical activity, presence of CVD, systolic blood pressure, BMI, and HbA1C	Multivariable logistic regression	Moderate vs. abstainers, OR: 0.60 (0.37–0.99), *p* = 0.023
Lee et al., 2010 Prospective n = 1239	9	Alcohol NS	Sex, age, ethnicity, smoking, HbA1c, BMI, systolic blood pressure, and duration diabetes	Multivariable logistic regression	Moderate vs. none, OR: 1.08 (0.70–1.67) Heavy vs. none, OR: 1.07 (0.54–2.13), *p* = 0.8
Moss et al., 1993 Prospective Younger: 439 Older: 478	9	Alcohol NS	Sex, age, HbA1c	Multivariable logistic regression	Younger-onset diabetics per 1oz/day increase in alcohol consumption on DR incidence, OR: 2.09 (0.04–1.07);per 1oz/day increase in alcohol consumption on DR progression, OR: 1.25 (0.75–2.08). Older-onset diabetics per 1oz/day increase in alcohol consumption on DR incidence, OR: 0.75 (0.4–1.42); per 1oz/day increase in alcohol consumption on DR progression, OR: 0.73 (0.4–1.20)
Moss et al., 1992 Cross-sectional Younger: 891 Older: 987	9	Alcohol Protective	Diabetes duration, age, HbA1c, diastolic blood pressure, insulin therapy	Multivariable logistic regression	Younger-onset diabetes population per 1oz/day increase in alcohol consumption for PDR, OR: 0.49, (0.27–0.92) Older-onset: no significant associations
Gupta et al., 2020 Prospective n = 656	8	Alcohol Protective	Age, sex, BMI, smoking, systolic blood pressure, income, HbA1c, diabetes duration, hyperlipidaemia, CKD, antidiabetic medication	Multivariable analyses	Alcohol consumption vs. non-drinkers, OR: 0.36 (0.13 to 0.98) *p* = 0.045; occasional drinker (≤2 days/week) vs. non-drinkers, OR:0.17, (0.04–0.69), *p* = 0.013)
Thapa et al., 2018 Cross-sectional n = 1860	8	Alcohol Risk	NIL	Multivariable logistic regression analysis	Alcohol consumption yes vs. no for DR (OR:4.3, 95%CI: 1.6–11.3, *p* = 0.004) and vision-threatening DR (OR: 8.6, 95%CI: 1.7–47.2, *p* = 0.010)
Harjutsalo et al., 2013 Cross-sectional n = 3608	8	Alcohol Protective	Sex, diabetes duration, age at onset of diabetes, triglycerides, HbA1C, HDL cholesterol, social class, BMI, smoking status, lipid-lowering agents and hypertension	Multivariable logistic regression	Abstainers vs. light consumers, OR: 1.42 (1.11–1.82), *p* < 0.05; former users vs. light consumers, OR: 1.73 (1.07–2.79), *p* < 0.05
Cundiff et al., 2005 Prospective n = 1412	8	Alcohol NS	Intake of energy	Spearman correlation	No significant association with DR (*p* = 0.26)
Young et al., 1984 Prospective n = 296	8	Alcohol Risk	Diabetes duration, impotence and glycemic control	Multivariable logistic regression	Heavy consumption vs. none–moderate consumption, RR: 2.25 (1.15–4.42)
Giuffre et al., 2004 Case–control Case:45 Ctrl:87	7	Alcohol NS	Diabetes duration, duration of oral treatment and duration of insulin therapy	Multivariable logistic regression	No significant association with DR (data not shown)
Kawasaki et al., 2018 Cross-sectional n = 363	5	Alcohol NS	Age, sex, HbA1c, diabetes duration, medication, BMI, lifetime maximum body weight, systolic blood pressure, diastolic blood pressure, non-HDL cholesterol, HDL-cholesterol, LDL, estimated glomerular filtration rate, history of myocardial infarction, history of stroke, alcohol, smoking, number of oral hypoglycemic agents, number of antihypertensive agents	Multiple logistic regression model	No signification was seen (*p* = 0.759)
Acan et al., 2018 Cross-sectional n = 413	3	Alcohol Risk	NIL	*t* test	*p* = 0.010
Mediterranean Diet					
Ghaemi et al., 2021 Prospective n = 22187	7	Mediterranean diet Protective	Age, sex, time, HbA1c, fasting plasma glucose, HDL-cholesterol, total cholesterol, total triglycerides, systolic blood pressure, obesity, smoking, diabetes duration	Pooled logistic regression models	Mediterranean diet against incident retinopathy in type 1 DM (OR: 0.32, 95%CI: 0.24–0.44, *p* = <0.001) and type 2 DM (OR: 0.68, 95%CI: 0.61–0.71, *p* = <0.001)
Diaz-Lopez et al., 2015 Interventional n = 3614	ModerateBias	Mediterranean diet Protective	Sex, age, waist circumference, BMI, smoking, physical activity, hypertension, educational level, dyslipidemia, family history of premature coronary heart disease, and baseline adherence	Multivariate Cox regression	Mediterranean diet vs. control diet, HR: 0.60 (0.37–0.96)
Caloric Intake					
Alcubierre et al., 2016 Case–control Case:146 Control:148	10	Caloric intake NS	Sex, age, diabetes duration, energy intake, systolic blood pressure, physical activity, waist circumference, HDL cholesterol, educational level and diabetes treatment	Multivariable logistic regression	Highest energy intake tertile (T3) vs. lowest energy intake tertile (T1), OR: 0.73 (0.37–1.46)
Roy et al., 2010 Prospective n = 469	10	Caloric intake Risk	Sex, age, total caloric intake, oleic acid intake, physical exercise, glycated hemoglobin, carbohydrate intake, protein intake, and hypertension	Multivariable logistic regression	Higher caloric intake, OR: 1.48 (1.15–1.92), *p* = 0.003
Cundiff et al., 2005 Prospective n = 1412	8	Caloric intake Risk	NIL	Spearman correlation	Calories in kcal against DR progression rate, r = 0.07 (*p* = 0.007)

BMI—Body mass index, CVD—Cardiovascular disease, CKD—Chronic kidney disease, DM—Diabetes Mellitus, DR—Diabetic retinopathy, HDL—High-density lipoprotein, HbA1c—Glycated hemoglobin, LDL—Low-density lipoprotein, OR—Odds ratio, PDR—Proliferative diabetic retinopathy, RR—Relative risk, VTDR—Vision-threatening diabetic retinopathy.

## Data Availability

Not applicable.

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
