# Peer review of "Dietary Intake and Diabetic Retinopathy: A Systematic Review of the Literature"

_nutrients, 2022, doi:10.3390/nu14235021_

Round 1
Reviewer 1 Report
It is very interesting paper, a very comprehensive description of the research on diet and DR. Diabetes is a very common chronic disease. As the most common microvascular complication of diabetes, there is no effective treatment in clinical practice at present, so diet management is very important for DR. This systematic review is very valuable and can provide a basis for ophthalmologists and endocrinologists to guide the diet of patients with diabetes or DR. I would recommend the article can be accepted with minor revision.
Minor comments:
1. The studies included in the review were based on different criteria for the diagnosis or staging of DR, including ETDRS and International Classification System. It is suggested that the authors briefly introduce the different diagnostic or staging criteria of DR In the methods section.
2. The article addressed DR and DME in the abstract and conclusion, but did not discuss the relationship between dietary intake and DME in the main body (Results). It is strongly suggested to carefully summarize the content in the article.
3. It is suggested to add Exclusion criteria for special types of diabetes, such as gestational diabetes.
4. I would suggest that the PRISMA Checklist should be added to the supplementary materials of the review to make this review more complete. PRISMA (prisma-statement.org)
Author Response
We would like to thank the Reviewer for taking his/her precious time and patience to review the manuscript. We sincerely appreciate all your valuable comments and suggestions for helping us in improving the quality of the manuscript. We address all the concerns here and amended the manuscript highlighted in red for reviewer 1.
Minor comments:
Point 1. The studies included in the review were based on different criteria for the diagnosis or staging of DR, including ETDRS and the International Classification System. It is suggested that the authors briefly introduce the different diagnostic or staging criteria of DR In the methods section.
Response 1: Thank you for the suggestion. A brief introduction about the different diagnosis or staging criteria of DR is included in the 4th point of inclusion criteria in the method section on page 4. “Studies that assessed DR outcomes by fundus photography, fundus examination using a direct or indirect ophthalmoscope, and fundus fluorescein angiography were accepted. Different scales for grading the severity of DR such as the Early Treatment Diabetic Retinopathy Study (ETDRS) and the International Classification system of DR were also accepted. The ETDRS is based on seven field stereophotograph classifying DR from level 10 (absence of retinopathy) to level 85 (vitreous hemorrhage or retinal detachment involving macula) whereas the International Classification System grade into no apparent retinopathy, mild, moderate, severe non-proliferative retinopathy, and final stage proliferative diabetic retinopathy [29].”
Point 2. The article addressed DR and DME in the abstract and conclusion but did not discuss the relationship between dietary intake and DME in the main body (Results). It is strongly suggested to carefully summarize the content in the article.
Response 2: Thank you for these observations. In our review, only one study was found to assess the influence of diet on DME. This point had been addressed in the results under section 3.4.6 sodium of micronutrients (page 20). This prospective study found to worsen DME with increased intake of sodium. As per your suggestion, we have now included this information in the abstract (page 1) and in the first paragraph of the discussion (page 33).
Abstract – “Only one study in our review assessed dietary intake and DME and found the risk of high sodium intake for DME progression.”
Discussion – “The assessment of the influence of dietary intake on DME is very limited and in our review, only one prospective study was found to evaluate it. This study found that high intake of sodium was associated with DME progression.”
- It is suggested to add Exclusion criteriafor special types of diabetes, such as gestational diabetes.
Response 3: We agree with the reviewer and have added a special type of diabetes such as gestational diabetes to the exclusion criteria.
- I would suggest that the PRISMA Checklist should be added to the supplementary materials of the review to make this review more complete. PRISMA (prisma-statement.org)
Response 4: Thank you for the suggestion. We have prepared the PRISMA checklist and included it in the supplementary material.

Reviewer 2 Report
This review is quite comprehensive and features citations that are relatively recent. There were a few areas where the grammar could be improved, but generally the precision of the review is quite good.
Author Response
We would like to thank the Reviewer for taking his/her precious time and patience to review the manuscript. We sincerely appreciate all your valuable comments and suggestions for helping us in improving the quality of the manuscript. We address all the concerns here and amended the manuscript highlighted in blue for Reviewer 2.
Reviewer 2
This review is quite comprehensive and features citations that are relatively recent. There were a few areas where the grammar could be improved, but generally the precision of the review is quite good.
We thank the Reviewer for this suggestion. We have now carefully looked through the manuscript and the changes are reflected within the manuscript in blue.
